# Loss of Bone Density in Patients with Anorexia Nervosa Food That Alone Will Not Cure

**DOI:** 10.3390/nu16213593

**Published:** 2024-10-23

**Authors:** Dennis Gibson, Zoe Filan, Patricia Westmoreland, Philip S. Mehler

**Affiliations:** 1ACUTE Center for Eating Disorders, Denver Health, Denver, CO 80204, USA; dennis.gibson@dhha.org (D.G.); zoe.filan@dhha.org (Z.F.); patricia.westmoreland@dhha.org (P.W.); 2Department of Medicine, University of Colorado, Denver, CO 80045, USA; 3Eating Recovery Center, Denver, CO 80230, USA

**Keywords:** osteoporosis, anorexia nervosa, malnutrition

## Abstract

Background: Anorexia Nervosa is a highly lethal illness that is also associated with many medical complications. Food restriction and weight loss define this illness. Most of its physical complications are reversible with weight restoration, with the notable exception of the loss of bone density, which is commonly present in anorexia nervosa. Methods: A comprehensive scientific literature review was performed in order to explore bone disease in anorexia nervosa. Results: The pathophysiology of the loss of bone mineral density in anorexia nervosa was elucidated, along with the diagnosis and treatment of osteoporosis in patients with anorexia nervosa, including the nutritional approach to weight restoration. Conclusions: Loss of bone mineral density in anorexia nervosa is very aggressive. Nutritional rehabilitation is a cornerstone to treating this, along with medicinal considerations.

## 1. Introduction

Anorexia nervosa (AN) is a psychiatric illness that leads to weight loss and malnutrition. Apart from deaths due to the opiate epidemic, it has the highest mortality of any psychiatric illness [1], with deaths being due to medical complications of the disorder or suicide [2]. AN has no single cause and is highly comorbid with other psychiatric disorders, including anxiety, depression, obsessive–compulsive disorder, and substance use disorders. Familial dysfunction; change in environment; emotional, sexual, or physical abuse; and peer pressure are common perceived triggers for anorexia nervosa. A genetic predisposition (generally toward an anxious and perfectionistic disposition) is necessary but not sufficient for the development of the disorder. Several neurobiological alterations have been found to be related to AN. Reductions in dopamine in individuals with eating disorders have been postulated to induce feelings of anxiety and unpleasantness during eating rather than the feelings of comfort in individuals who do not have eating disorders. According to Higgins et al., this may explain why individuals with AN report food-elicited sensations as aversive rather than satisfying [3]. Serotoninergic systems appear to be altered as well and may mediate many of the comorbidities that have been associated with AN, such as depression, obsessive–compulsive disorder, and anxiety. Significant genetic correlations have been found among AN and various anthropometric and metabolic traits, such as body mass index (BMI) and fasting blood glucose [4]. The lifetime prevalence of AN is 0.9% among women with a 20:1 female-to-male ratio.

Individuals with a restricting type of AN (AN-R) lose weight by limiting their intake, and while individuals with binge–purging-type anorexia (AN-BP) may restrict their intake, they also engage in compensatory purging behaviors (e.g., vomiting and laxative use) to rid their bodies of calories. Morbidity in AN-R is related to the extent of malnutrition-induced weight loss, whereas morbidity in individuals with the purging subtype is related to the mode and frequency of purging [5]. Due to societal-based weight norms and pressures, individuals who are thin are often lauded for their appearance until they reach markedly abnormal weights. In addition, medical providers typically receive minimal training in the detection or treatment of eating disorders [6]. Even if individuals with AN are identified and referred for treatment, treatment is costly and often not available to individuals, especially males, without insurance. In addition, there are no FDA-approved medications for the treatment of AN. The course of adolescent-onset anorexia has shown a favorable outcome, with a 30-year follow-up study indicating that most patients with this disorder do well eventually [7]. However, there are multiple obstacles to successful treatment, and many individuals with AN have a protracted course of illness and a reported relapse rate of around 25% [2]. Repeated treatment failures, insurance denials leading to premature discharge, revolving-door experiences, and treating clinician fatigue potentially contribute to a worrisome pattern of opting for palliative care or hospice for the chronically ill [8].

Thankfully, despite many problems inherent to AN, almost all of the medical complications that result from AN are reversible with appropriate nutrition and weight restoration [5]. The exception is the deleterious effect of malnutrition on bone mineral density (BMD), such as loss of height, chronic pain, and immediate and long-term risk of fracture, which are not completely reversible despite weight restoration [4]. Factors associated with an increased risk of low BMD include lower BMI, longer duration of illness and amenorrhea, decreased muscle mass, and lower serum vitamin D levels [9,10]. In this review paper, we discuss the causes of low bone density in AN, its diagnosis and treatment, along with the role of nutritional rehabilitation as a treatment modality.

## 2. Pathophysiology of Low Bone Density

The pathophysiology underlying osteoporosis and osteopenia in patients with AN is identical, with the difference being that the loss of BMD is less severe in osteopenia. The surprising issue remains the aforementioned high frequency of these adverse conditions in AN, notwithstanding the young age of these patients. Typically, osteoporosis and osteopenia are mostly found in older males and postmenopausal females. In order to better understand the unexpectedly high prevalence of loss of BMD in patients with AN, one must briefly review the etiology of osteoporosis. Specifically, there are two main bone cell types that determine normal BMD, namely, osteoclasts and osteoblasts. Bone in the human body is constantly in a state of resorption and formation [11]. During adolescence and adulthood, there is more bone formation than resorption occurring. Thus, the activity of the osteoblasts, which are responsible for new bone formation, exceeds that of the osteoclasts, which cause bone resorption in younger people. As normal adults reach the ages of 50–60, bone resorptive activity increases more, causing a slow net decrease in BMD and the potential beginning of osteopenia and osteoporosis. However, in AN, there is an “uncoupling” phenomenon that can begin as early as the teenage years, which results in increased bone resorption along with decreased bone formation [12]. This premature “double whammy” is what causes the development of serious loss of BMD as early as the late teens and early twenties, whereas in normal adults, the occurrence of decreased bone formation is delayed for decades. In AN, low levels of insulin-like growth factor-1 (IGF-1) reduce the serum levels of osteocalcin, a protein that reflects bone formation, and cause abnormalities in the osteoblast with resultant decreased bone formation, notwithstanding their usual young ages. In addition, AN is characterized by elevated serum levels of cortisol, which is inversely related to levels of osteocalcin [13].

There are also many other factors in AN that result in this premature loss of BMD. These include (1) decreased intake of calcium, phosphorus, and vitamin D; (2) diets low in animal protein; (3) excessive intake of carbonated soda drinks in a subcategory of patients; (4) compulsive weight-bearing exercises; (5) low levels of sex hormones (amenorrhea); (6) depleted fat stores; (7) low leptin levels; and (8) overall malnutrition [9]. Adipose tissue is required to reverse the endocrinopathies that favor bone catabolism by causing an increase in serum leptin level, which in turn results in normalization of pituitary and gonadal sex hormone secretion [9]. It is the confluence of these unique features of AN, the aberrations in the endocrine system due to AN, and the overall state of caloric deprivation that results in the bone’s low turnover state characterized by both increased bone resorption and decreased formation, which in turn causes the commonly diagnosed low BMD found in patients with AN [14].

## 3. Evaluation of Bone Density

Guidelines for the workup and treatment of osteoporosis are largely based on data from the postmenopausal population, creating significant knowledge gaps regarding bone health in those with AN, a majority of whom are premenopausal females. Although dual-energy X-ray absorptiometry (DXA) T-scores of −1 to −2.5 of the lumbar spine, total hip, or femoral neck are utilized for a diagnosis of osteopenia and −2.5 or less for a diagnosis of osteoporosis in postmenopausal individuals, Z-scores are utilized for premenopausal individuals, with a Z-score cutoff of −2.0 or less being defined as “below the expected range for age”, as the term osteopenia should be avoided in this premenopausal population. Thus, scores greater than −2.0 are considered normal bone mineral density (BMD). Males under the age of 50 years should not be diagnosed with osteoporosis based on BMD alone. Similarly, a Z-score less than −2.0 does not automatically confer use of the term osteoporosis in the premenopausal population, but a Z-score less than −2.0 along with any condition associated with increased bone loss, such as AN, or a history of a fragility fracture with any BMD allow for a diagnosis of osteoporosis [15]; hence, all individuals with AN and a Z-score less than −2.0 meet criteria for osteoporosis given the effects of starvation on bone health.

Recommendations on timing of initial and repeat BMD testing (DXA) in the premenopausal AN population are unclear, with current guidelines suggesting follow-up BMD testing when the results are likely to influence management and the determination of intervals between BMD testing dependent on each patient’s clinical status [15]. Expert opinion recommends an initial DXA for any individual with active AN and/or amenorrhea for more than 6–12 months [16].

Regarding transgender and gender nonconforming individuals, the International Society for Clinical Densitometry (ICSD) recommends a baseline DEXA for any patient with a history of gonadectomy or therapy that lowers endogenous sex hormone production. Then, repeat BMD testing is warranted every 1–2 years thereafter until BMD is stable or improved. Z-scores based on the gender identity of the individuals should be used; for gender non-binary individuals, a Z-score based on the sex recorded at birth is recommended [15].

Based on the previously discussed diagnostic criteria, Frølich et al. (2020) found that 50% of adult women with active AN have Z-scores less than −2.0 on DXA [17]. Schorr et al. (2019) found 65% of men with AN meet the criteria for osteoporosis [9]. However, the prevalence of osteoporosis is found at much lower rates in children and young adolescents with AN. Specifically, 11–18.9% of young adolescent girls with a mean age of 16 years met the DXA criteria for osteoporosis [18,19]. Similar studies using a Z-score cutoff of −2.0 have not been completed in adolescent boys, although an increased rate of negative Z-scores is still found in this cohort [20]. Nonetheless, long-term fracture risk seems to increase. Fracture risk for females with AN remains increased at all ages [21], but studies on males are divergent, with one study finding no increased risk [22] and a different study finding increased risk after the age of 40 years [21]. It also seems that patients with AN-R are at higher risk for low BMD than individuals with AN-BP [10].

Qualitative markers of bone health, including various laboratory investigations associated with bone formation and resorption, allow for a more dynamic assessment of bone turnover and response to therapeutic interventions, even before radiologic changes may be noticed. These studies further support the deleterious effects of starvation and hormonal changes on bone health in this population, finding reduced bone formation and resorption in the premenopausal population, as evidenced by reduced bone markers for each process [23]. Haschka et al. (2014) found higher cross-linked C-telopeptide, a marker of bone resorption, in young adult females with AN and low BMD [24]. Wanby et al. (2021) similarly found increased levels of osteopontin, a marker of bone resorption, in adult females with AN and reduced bone mineral density [25]. Oświęcimska et al. (2007) found a significant increase in bone alkaline phosphatase, a marker of bone formation, in adolescent girls with AN as body mass index (BMI) increased with refeeding over a 19-month period, although Z-scores on DXA did not significantly change during this same time period [26].

Although DXA remains the preferred method of testing for BMD and diagnosing osteoporosis, it does not actually measure bone quality or strength, and its utility in this population remains unclear due to the poor correlation with fracture risk [17,27]. Bone strength is determined by several properties: BMD, bone composition, bone geometry, and bone microarchitecture. Trabecular bone score (TBS) has been examined as a potential independent addition to BMD, with lower scores indicating weaker bone microarchitecture. Studies on adults and adolescents with AN suggest a positive correlation between TBS and lumbar spine BMD, although most patients do not have significantly compromised microarchitecture, even with low BMD [24,28,29]. However, the ability of TBS to predict fracture risk when combined with DXA in the premenopausal AN cohort remains unclear, and how best to utilize TBS in this population to help guide therapy also remains poorly defined. Similarly, peripheral DXA utilizing the distal radius of the non-dominant forearm should not be utilized as an alternative to central DXA at this time.

Quantitative computed tomography (QCT) and peripheral QCT are additional methods to assess bone health, providing a better representation of bone geometry than the two-dimensional imaging used with DXA, as the QCT utilizes three-dimensional imaging to measure volumetric bone and can also separately assess cortical and trabecular bones. A positive correlation between trabecular volumetric BMD and DXA BMD in adolescents with AN has been found [30]. However, the benefits of QCT versus DXA are unclear. QCT is also associated with a significantly increased risk of radiation and does not measure hip or spine bone density. Quantitative ultrasound (QUS) is another method to assess bone health that correlates with hip and spine BMD in adult women with AN without associated radiation, but, again, the benefit of this technology toward assessing and treating bone health in this population remains unclear [31].

## 4. Treatment of Low Bone Mineral Density in AN

Weight restoration, ideally with resultant normalization of hypothalamic–pituitary–gonadal function and resumption of menses, is a critical component of treatment for low BMD in AN [32]. In a systematic review by El Ghoch et al. (2016), female adolescents experienced stabilization of BMD at one year of follow-up after restoration of body weight [33]. Thereafter, a 3–4% yearly rate of increase in BMD is expected with continued weight recovery [15]. It is unlikely, however, that complete normalization of BMD can be achieved through weight recovery alone [34], although one study in female adolescents found normalization after 30 months of weight restoration and resumption of menses [35]. The impact of weight gain itself versus normalization of the gonadal hormone axis toward bone health remains unclear, as both independently seem to favorably impact BMD [15].

Adequate intake of calcium and vitamin D, with daily ingestion of 1,200 mg of calcium and 800 international units (IU) of vitamin D, from all sources, is also warranted for patients with lower BMD, especially given the often-found reduced levels of serum vitamin D in individuals with AN [36,37]. Inadequate serum 25-hydroxy vitamin D (<30 ng/mL) also attenuates the improvement in BMD seen with weight restoration [38]. However, as mentioned above, advanced osteoporosis is unlikely to be reversed solely with nutritional factors, and medications are often required [34]. Prior to the late 1990s, no medications were available to treat the loss of BMD found in AN. Currently, there are several different classes of medications with likely efficacy. Of note, medical therapy is only indicated in osteoporosis but not in osteopenia, where treatment involves nutritional rehabilitation and weight restoration.

### 4.1. Bisphosphonates

Bisphosphonates are a class of medication that impair bone resorption through inhibition of osteoclasts. Studies utilizing bisphosphonates have revealed conflicting results. Administration of risedronate to adult women with AN resulted in either increased spinal BMD or an increase in both hip and spine BMD [39,40]; however, administration of alendronate to adolescents with AN failed to increase BMD in comparison to the placebo group [41]. These findings are likely related to the differences in bone turnover between adolescents and adults with AN—adolescents with AN have reduced bone formation and resorption, while adults with AN have reduced bone formation but increased resorption [42]. Also, it remains unclear if bisphosphonates actually reduce fracture risk in the AN population, as atypical long bone fractures, which are low-trauma femoral shaft fractures with unusual radiographic features, occur with bisphosphonate use in the non-eating disorder population. Other adverse effects of bisphosphonates can include more commonly reported upper gastrointestinal intolerances and jaw osteonecrosis, a rare but serious complication associated with this class of medications [43]. There is also a theoretical risk to neonatal health during pregnancy with recent use of bisphosphonates due to the long-lasting accumulation of bisphosphonates in maternal bones [44]. Overall, many clinicians favor judicious use of bisphosphonates in adult AN, including weekly oral alendronate as well as intravenous once-yearly zoledronic acid.

### 4.2. Hormonal Therapy

Estrogen is critical in the maintenance of bone density, maintaining balance between bone formation and resorption, and reducing overall bone resorption [45]. However, only transdermal (TD) estrogen has shown benefit in increasing bone density in both adolescents and adults with AN, while the estrogen found in oral contraceptives [46,47,48] is efficacious in postmenopausal osteoporosis. This is likely related to the divergent effects on insulin-like growth factor 1 (IGF-1), the anabolic hormone that mediates the effects of growth hormone. Oral estrogen is first metabolized in the liver, where it acts to suppress IGF-1 production and secretion from the liver; however, TD estrogen does not have the same effects involving the liver due to its cutaneous absorption [49]. This is supported by the findings from Grinspoon et al. (2002), who found that the combination of oral estrogen and recombinant human IGF-1 had the greatest increase in BMD, while the addition of recombinant human IGF-1 to TD estrogen resulted in no additional benefit to bone health [50,51].

The major function of testosterone on BMD is also through reduced bone resorption, but much of this is mediated through the effects of estrogen from peripheral aromatization of testosterone [45]. However, the use of TD testosterone in women with AN does not seem to improve BMD based on a single study [41]. The effects of testosterone on BMD in males with AN have not been previously studied. However, TD testosterone is frequently recommended by these authors for males who have lower serum testosterone levels in the form of intramuscular testosterone to achieve a serum level of approximately 500 ng/dL. When testosterone is being administered, there should be a trial of withholding it to recheck serum levels once satisfactory weight restoration has occurred because it is likely that endogenous production of testosterone will resume and deem the supplementation unnecessary. Closure of the epiphyseal plates should be verified in adolescent males with wrist X-rays prior to initiation of testosterone.

Limited research supports the use of IGF-1 in the treatment of bone disease in AN. One study showed an increase in spinal BMD in adult women after receiving recombinant human IGF-1 versus placebo for nine months (*p* = 0.05) [52]. Another study showed that sequential treatment of low BMD with recombinant human IGF-1 followed by risedronate (bisphosphonate) increased spine BMD in adult women with AN more than bisphosphonate alone [51]. Recombinant human IGF-1 administration to adolescent females with AN for a short period of time also caused an increase in surrogate markers of bone formation, although BMD itself was not studied [53].

Several other studies have also examined the impact of dehydroepiandrosterone (DHEA), a steroid androgenic hormone precursor, on BMD, but DHEA is not recommended for the treatment of bone health in individuals with AN [54]. Serum leptin levels are extremely low in AN, but treatment with recombinant human leptin is also not currently recommended for the treatment of low BMD in people with AN, as it is not currently well supported. In one small pilot study (*n* = 20), two years of treatment with leptin did improve BMD in strenuously exercising young women with hypothalamic amenorrhea, although they did not have a diagnosed eating disorder [55].

### 4.3. Denosumab

Denosumab is a monoclonal antibody that inhibits bone resorption through prevention of osteoclast activity. Several case reports support the use of this medication to improve BMD in those with AN. Jamieson and Pelosi (2016) reported a 14.8% improvement in BMD at the lumbar spine, 1.4% at the left total hip, and 5.7% at the left femoral neck after three years of therapy in a 29-year-old woman [56]. Isobe et al. (2018) described three adult female patients with AN and osteoporosis who tolerated well 24 months of therapy with denosumab and demonstrated reduced levels of surrogate markers of bone turnover as well as increased spine and hip BMD [57]. Finally, Kilbane et al. (2020) described a 21.6% increase in spine BMD and a 28.6% increase in total hip BMD following two years of denosumab in a 51-year-old postmenopausal female with AN [58]. A randomized clinical trial of 30 women with AN with an average age of 29 years and a BMI of 19 kg/m^2^ also recorded a significant increase in spine BMD compared to the placebo group with 12 months of therapy [59]. Denosumab also seems to be a reasonable choice for adults with AN and low BMD but has not been studied in adolescents. Adverse events can include increased risk of fracture with discontinuation of therapy, hypocalcemia, jaw osteonecrosis (extremely rare), and increased risk of fetal malformation due to its teratogenicity (pregnancy category X) [44].

### 4.4. Teriparatide

Teriparatide (recombinant human parathyroid hormone) is the only currently used anabolic agent that increases bone formation. It does so by stimulating osteoblast function versus other medication classes that decrease bone resorption. One case report described a 52-year-old woman with AN who experienced a 21% increase in bone density over 2 years of treatment with teriparatide, along with resolution of fractures [60]. Another case report found complete normalization of spinal and hip BMD in a 27-year-old woman with osteoporosis, although other medications were utilized [61]. Similarly, Fazeli et al. (2014) found a significant increase in spine BMD in a randomized, placebo-controlled trial after only 6 months of teriparatide therapy in adult women with AN [62]. This medication thus seems effective for adults with AN and low BMD but has not been studied in adolescents. However, this medication should be avoided in any patient with an increased risk of osteosarcoma due to its anabolic effects on bone formation.

### 4.5. Exercise

Weight-bearing exercise normally has a salutary effect in enhancing bone strength by imparting mechanical loads on the bone, which is certainly true in postmenopausal osteoporosis. However, the effects of physical activity on BMD in people with AN are unclear, as much of the research is older and of lower quality. Generally, these authors recommend that excessive weight-bearing exercise or any moderate-to-high-intensity exercise be avoided until weight is restored and menses are resumed because of the potential deleterious effects on the microarchitecture of bones. Indeed, Waugh et al. (2011) reported that excessive exercise with moderate load on the bone resulted in further bone density loss in people with active AN [63]. These adverse effects of exercise on bone health in active AN are likely mediated by estrogen deficiency [64]. Of note, much of this discussion is relevant to low energy availability (REDs) in sports [65].

## 5. Approach to Refeeding a Patient with AN

Nutritional rehabilitation, or the process of restoring or optimizing nutritional status, is essential in the treatment of AN. While a “start low, advance slow” approach to refeeding patients with AN was thought to be best practice for many years, recent studies support the safety and efficacy of a more aggressive approach [66]. When beginning the inpatient refeeding process, a registered dietitian (RD) may prescribe a patient with AN a meal plan of between 1400 and 1800 calories per day to start. Where in this range the initial calorie goal falls will depend on the patient’s age, their sex, as well as how far they are from their healthy or “ideal” body weight (IBW). For example, adolescents, adult males, and those at or above 85 percent of their IBW may start closer to the 1800 calorie goal. From there, the RD will increase calories by 400 to 500 calories as often as every two to three days or until a consistent weight gain of approximately 0.2 kg per day is observed. Because the goal of refeeding in patients with AN is to support ongoing weight gain toward a healthy body weight, a maximum calorie goal is rarely set [67]. Rather, the RD will monitor weight trends throughout the process and adjust calories as needed for weight restoration while monitoring liver transaminase levels.

The RD may also calculate a patient’s basal energy expenditure (BEE) using the Harris–Benedict equation (although indirect calorimetry is preferred, if available) and multiply this total by a stress factor of 2.0–2.5 to determine a critically underweight patient’s total energy expenditure (TEE). This can be helpful when estimating the daily calorie target that may need to be achieved days to weeks after the initiation of refeeding for patients with AN.

Most RDs agree that it is helpful to allow patients to choose their own food during the refeeding process. Typically, the macronutrient composition followed is 40% of calories from carbohydrates, 40% of calories from fat, and 20% of calories from protein to help condense menus and manage the feeling of fullness that is often seen with weight restoration meal plans. Normalization of menus and the introduction of a broad variety of foods, including solid foods, is encouraged once the patient’s condition is medically appropriate. Some AN patients may require the initiation of enteral or, rarely, parenteral nutrition if they are unable to ingest adequate nutrition by mouth. Enteral feeding can be a very useful tool for patients suffering from the physiological impediments to eating by mouth that can be associated with AN, such as superior mesenteric artery (SMA) syndrome, dysphagia, and gastroparesis [68].

A very important consideration when refeeding patients with AN is the potentially deadly complication that can occur during this process: refeeding syndrome (RS). RS is historically described as “a range of metabolic and electrolyte alterations occurring as a result of the reintroduction and/or increased provision of calories after a period of decreased or absent caloric intake” [69]. Patients with AN are generally at increased risk for RS, not only because they will begin treatment and refeeding after long periods of energy restriction but also because they are likely to have experienced recent weight loss resulting in low BMI. For this reason, it is of utmost importance that patients with severe AN (BMI < 14 kg/mg^2^) are initially refed during an inpatient hospitalization. This way, the treatment team can employ strategies to prevent RS, such as measuring and correcting serum electrolyte levels before initiating refeeding and monitoring these levels daily for at least 1–2 weeks afterward with timely supplementation of phosphorus and potassium, as dictated by serum levels.

## 6. Conclusions

Low BMD is a surprisingly common and serious complication of AN. It thus behooves dieticians and nutritionists to have some familiarity with this problem so that they too can discuss it with patients who have AN. While weight restoration is very important for overall recovery from AN, other additional treatments should be considered for low BMD, which remains a potentially irreversible complication if not specifically diagnosed timely and treated effectively.

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
