# Peer review of "Loss of Bone Density in Patients with Anorexia Nervosa Food That Alone Will Not Cure"

_nutrients, 2024, doi:10.3390/nu16213593_

Round 1
Reviewer 1 Report
Comments and Suggestions for Authors
I have the following suggestions for revision:
I think the abstract can be more informative about what you specifically found in the review. I suggest highlighting your main findings in the abstract, especially the nutritional findings.
Line 30 alterations in dopamine should this be reductions in dopamine?
Line 87 it would be of benefit if you could explain or define osteocalcin here.
Line 104 in this paragraph I think it would be beneficial to provide definitions of T scores versus Z scores. To determine one’s risk of fracture, there are tools that are based on bone density and other characteristics of the patient, for example, the FRAX tool. This could be mentioned in this paragraph.
Line 167 change “the ability to” to “the ability of”
Line 172 in this paragraph you could mention the drawback of PQCT or ultrasound in that they cannot measure bone at clinically relevant sites, such as the hip or spine.
Line 190: “ It is unlikely, however, that complete normalization of BMD can be achieved through weight recovery alone” - a reference is needed to support this statement.
Line 200: “ However, as mentioned above, advanced osteoporosis is unlikely to be reversed singularly with nutritional factors, but rather medications are often required.” - again a reference would be beneficial here. I would suggest referencing recent osteoporosis guidelines.
Line 216: “… as atypical long bone fractures occur with bisphosphonate use in the non-eating disorder population” - define what is meant by an atypical fracture. Also, it would be beneficial to have a reference to support this statement.
Line 217 again a reference is needed to support this statement about this bisphosphonate side effects.
Line 280 again a reference is needed to support this statement on side effects
For the paragraphs starting lines 325 and 330: again these paragraphs are lacking any references to the literature to support these recommendations.
Comments on the Quality of English Language
The English is mostly fine
Author Response
Reviewer 1
- Line 30: Thank you. We have now changed “alterations” to “reductions” as suggested.
- Line 87: We have now defined osteocalcin as suggested.
- Line 104: We have currently explained the difference between T&Z scores in lines 106 – 118 and do not see the need to further define. Moreover, the FRAX Tool is not utilized in anorexia nervosa, presumably because it is mostly used in post-menopausal women.
- Line 167: “to” has been changed to “of” as suggested.
- Line 172: We have now added the additional limitation of QCT for spine and hip measurements as suggested.
- Line 190: A reference has been added as requested.
- Line 200: A reference has been added as suggested.
- Line 216: Atypical fractures have now been defined.
- Line 217: A reference has been added regarding the side effects.
- Line 280: A reference has been added as requested.
- Line 325 & 330: A reference has now been added.
Reviewer 2 Report
Comments and Suggestions for Authors
Dear authors,
Concerning my small suggestions, please open the attached file.
Kind regards

Author Response
Reviewer 2
Thank you for your kind comments.
- Line 10: “Physical” has been added as suggested.
- Line 26: “emotional” has been added as suggested.
- Line 41: “e.g.” has been added.
- Line 47: “especially males” has been added.
- Line 56: “clinician burnout,” has been replaced by “treating clinician fatigue,” to improve clarity as suggested.
- Line 93: “in a subgroup of patients” has been added.
- Line 260: The lack of credible support for leptin treatment is now mentioned.
Additional comment regarding the refeeding syndrome, we prefer not to add to the paper because it is not clear that the “speed of refeeding is causal in causing the refeeding syndrome.
Reviewer 3 Report
Comments and Suggestions for Authors
Review in the attachment.

Author Response
Reviewer 3
- We agree that the title could be made more precise and we prefer “Loss of Bone Density,” rather than “osteoporosis,” since we also discuss osteopenia.
- The abstract is by nature general since it is not a randomized trial nor a systematic review.
- Thank you for this suggestion. We have now added a sentence to define the paper’s objective.
- The applied literature review method is explained in the abstract.
- Epidemiological data has now been added in line 38.
- A sentence and a reference have been added regarding athletics and bone disease.
- Pathophysiology of Low Bone Density has been added to the 2nd section’s title. Thank you.
- We disagree with the concern about the content being a bit general. Rather, we were intentional in the scope of the manuscript since the intended audience is likely dieticians and other nutrition-relate professionals.
- We have expanded the conclusion as suggested.
- We did not include a limitations for this type of a review paper.
Round 2
Reviewer 3 Report
Comments and Suggestions for Authors
I accept the paper in this version.